# Boom-bust population dynamics increase diversity in evolving competitive communities

Michael Doebeli [1][✉], Eduardo Cancino Jaque [2] & Yaroslav Ispolatov[2]

The processes and mechanisms underlying the origin and maintenance of biological diversity have long been of central importance in ecology and evolution. The competitive exclusion principle states that the number of coexisting species is limited by the number of resources, or by the species' similarity in resource use. Natural systems such as the extreme diversity of unicellular life in the oceans provide counter examples. It is known that mathematical models incorporating population fluctuations can lead to violations of the exclusion principle. Here we use simple eco-evolutionary models to show that a certain type of population dynamics, boom-bust dynamics, can allow for the evolution of much larger amounts of diversity than would be expected with stable equilibrium dynamics. Boom-bust dynamics are characterized by long periods of almost exponential growth (boom) and a subsequent population crash due to competition (bust). When such ecological dynamics are incorporated into an evolutionary model that allows for adaptive diversification in continuous phenotype spaces, desynchronization of the boom-bust cycles of coexisting species can lead to the maintenance of high levels of diversity.

[1] Department of Zoology and Department of Mathematics, University of British Columbia, 6270 University Boulevard, Vancouver, BC, Canada. [2] Physics Department, University of Santiago of Chile (USACH), Santiago, Chile. [✉]email: doebeli@zoology.ubc.ca

The amazing diversity of life has sustained the debate about the origins and limits of biodiversity. While random, selectively neutral processes are thought by some to play an important role, e.g., in ecosystem dynamics[1] and in molecular evolution, it seems that a majority of researchers would agree that non-neutral ecological interactions—competition, predation, mutualism—are central to understanding diversity, with competition having received the most attention. Coexistence between competing species requires that intraspecific competition is strong enough relative to interspecific competition. This is captured by the concept of limiting similarity: to coexist, populations must be sufficiently different in their resource use. If populations use the same resource in the same way, they cannot coexist, a phenomenon known as the competitive exclusion principle[2].

The exclusion principle has faced challenges from many empirical counter examples, in which the number of coexisting and ecologically interacting species was significantly higher than the number of limiting resources. The best known such example is the Paradox of the Plankton[3], which is based on a comparison between the relatively small number of biochemical resources essential for plankton growth, and the number of known coexisting plankton species, which is orders of magnitude larger. Different theoretical explanations for conditions that circumvent the exclusion principle have been proposed, and it is known that fluctuating population sizes can lead to violations[2,4–6]. With fluctuating population sizes, the storage effect[7,8], as well as relative non-linearities[2] can lead to coexistence of more competitor species than essential resource species, e.g., because of cyclical dominance between competitors[9]. Most of these examples involve models with a finite number of distinct resources, but coexistence due to fluctuating population dynamics has also been shown in models with continuous niches. With continuous, externally imposed (seasonal) periodic cycles in population sizes, time essentially becomes an additional niche dimension along which populations can segregate and coexist[10]. In an evolutionary context, it has been shown that limiting similarity in a continuous niche space used by an evolving community whose member species are undergoing externally forced population fluctuations, larger amplitude fluctuations lead to more diversity, and hence effectively to smaller limiting similarity[11].

Most previous models used in his context have assumed that the population fluctuations are externally imposed, and that there is a finite number of distinct resources. Here we investigate the questions of evolving diversity and limiting similarity in a setting where population fluctuations are not externally imposed, but are instead due to competitive interactions within and between the evolving species. In addition, we address the question of diversity in continuous phenotypes spaces, corresponding to continuously varying resource use. Rather than the 1-dimensional phenotype spaces that are usually assumed with continuous resource distributions[11–13], our phenotype spaces are potentially high-dimensional.

We use the models of[14–18], which are extensions of classical competition models to high-dimensional continuous phenotype spaces. In previous work, we assumed that the underlying ecological dynamics have a stable equilibrium (the carrying capacity). However, by using difference equations rather than differential equations to describe ecological dynamics, it is straightforward to extend these models to allow for complicated ecological dynamics, which are by definition endogenously generated (i.e., the population fluctuations are a result of the competitive interactions). We show that for certain types of endogenously generated fluctuations, which we term "boom-bust" dynamics, the amount of diversity that evolves and is maintained at evolutionary steady state can be much larger than the diversity maintained without ecological fluctuations.

Boom-bust dynamics consist of long periods of (near-) exponential growth followed by a deep crash, in such a way that the boom-bust cycles of different species become spontaneously desynchronized. The amount of excess diversity enabled by boom-bust dynamics increases with the dimension of phenotype space, so that species can be much more tightly packed in high-dimensional spaces, corresponding to a much smaller limiting similarity necessary for coexistence in high-dimensional niche spaces.

Apart from asynchronous boom-bust dynamics, essentially all other types of complex fluctuations, including asynchronous chaotic dynamics not exhibiting the boom-bust fluctuations, do not increase the diversity at evolutionary steady. Our models thus provide a specific and robust mechanism for the evolutionary origin and maintenance of highly diverse competitive communities.

## Results

To accommodate various types of ecological dynamics, we consider ecological models given by difference equations, and hence set in discrete time. The basic ecological model we use is a difference equation[19–21] that links population densities $N$ of two consecutive generations $t$ and $t+1$,

$$N(t+1) = F(N(t)) = N(t) \frac{\lambda}{1 + aN(t)^\beta}. \qquad (1)$$

where $\lambda > 0$ is the per capita number of offspring, and $a > 0$ and $\beta > 0$ are parameters describing the effect of competition. For $\lambda < 1$, $N(t)$ converges to 0 for any initial condition $N(0) > 0$, and hence extinction is the only possible outcome. We therefore assume $\lambda > 1$ in what follows. In that case, model (1) has a non-zero equilibrium at $K = ((\lambda-1)/a)^{1/\beta}$, which is the carrying capacity of the population. It is then convenient to write (1) as:

$$N(t+1) = N(t) \frac{\lambda}{1 + (\lambda - 1)(N(t)/K)^\beta}, \qquad (2)$$

as this makes it easy to formulate the model in terms of continuous phenotypes (see below). Model (2) was shown to fit well a wide range of data[20], and for $\beta = 1$ can be derived from the logistic differential equation by integration over a finite time interval[22]. The model given by (1) and (2) is phenomenological in nature. Its basic dynamic properties are briefly described in the first section of Methods. While other simple discrete-time models can be derived from first principles, this does not appear to be the case for model (1) if $\beta > 1$[23]. Rather, this model should be viewed as a heuristic model that can exhibit a wide array of dynamic regimes, including the boom-bust regime that will be of paramount importance in this paper (see below).

Because the difference Eq. (2) has three rather than two parameters, it can exhibit certain dynamical properties that better known difference equations, such as the Ricker equation[20,24], do not have. For example, for small values of $\lambda$, model (2) can exhibit highly chaotic dynamics (as e.g., measured by the Lyapunov exponent) despite the fluctuations in population size being (arbitrarily) small (Fig. 1). Importantly, for small values of $\lambda$, and for large enough $\beta$, model (2) exhibits "boom-bust" dynamics (Fig. 1), in which long periods of near-exponential growth (due to small $\lambda$) are followed by deep crashes (due to high $\beta$) once the population size is above $K$ for the first time after the exponential phase. This cycle repeats itself qualitatively, but the dynamics is in fact chaotic and exhibits sensitive dependence on initial conditions, because the population size after the crash, and hence the length of the subsequent exponential phase, is different in each cycle. We note that such boom-bust dynamics cannot be observed in the Ricker model. Some of the possible dynamic regimes of model (2) and transitions between them are shown in Fig. 1.

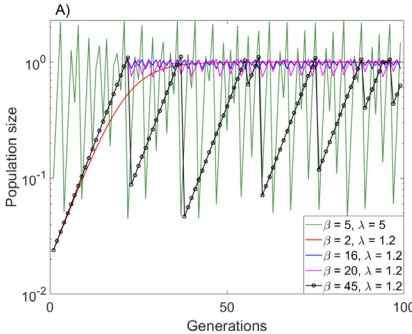
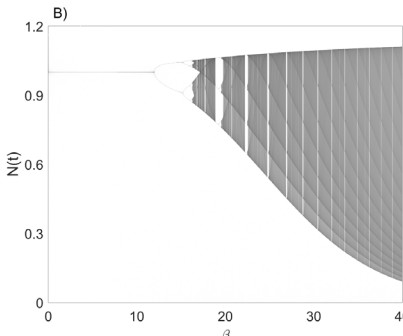

**Fig. 1 The dynamic complexity of the basic model. A** Examples of population dynamics generated by model (2). Shown are convergence to steady state for $\beta = 2, \lambda = 1.2$ (red line), stable periodic oscillations for $\beta = 16, \lambda = 1.2$ (blue line), chaotic dynamics when $\beta = 20, \lambda = 1.2$ (magenta line) and $\beta = 5, \lambda = 5$ (green line), and boom-bust chaotic dynamics when $\beta = 45, \lambda = 1.2$ (black line with individual generations shown by circles). Note the almost exponential multi-generation growth phases in the boom-bust regime, followed by a single-generation bust. **B** Bifurcation diagram for Eq. (2) for $\lambda = 1.2$. Both panels computed with $K = 1$.

We note that there are in principle many different models that can exhibit boom-bust dynamics (including models set in continuous time, see Discussion section). We chose model (2) as a generic model with boom-bust dynamics for certain parameter regions, viz. for $\lambda$-values close to 1 and large enough $\beta$. Rather than being interested in the likelihood of a particular model exhibiting boom-bust dynamics, we are interested in the consequences of such dynamics for the evolution of diversity. Therefore, while pointing out the contrast to the consequences of other types of ecological dynamics, such as cyclic or "regular" chaotic dynamics, delineating the different regions in parameter space generating the different types of dynamics is not relevant for our purposes.

We now consider a generalization of Eq. (2) that includes competition between $S$ phenotypically monomorphic populations. Each population is characterized by its phenotype $x_s = (x_s^1, ..., x_s^d) \in \mathbb{R}^d, s = 1, \ldots, S$, where $d$ is the dimension of phenotype space (which is assumed to be Euclidean $d$-space). The population size of phenotype $x_s$ is denoted by $N_s$. The ecological dynamics of all $S$ clusters are determined by the competition kernel $\alpha(x_s, x_r)$, which measures the competitive impact of phenotype $x_r$ on phenotype $x_s$, and the carrying capacity $K(x_s)$, which is the equilibrium population size of phenotype $x_s$ in the absence of any other phenotypes. (The competition kernel and the carrying capacity are functions $\alpha : \mathbb{R}^d \times \mathbb{R}^d \to \mathbb{R}$ and $K : \mathbb{R}^d \to \mathbb{R}$, respectively.)

The discrete time dynamics of each phenotype in the competitive community is then given by

$$N_s(t+1) = N_s(t)$$
$$\times \frac{\lambda}{1 + (\lambda - 1)\left[\sum_{p=1}^{S} N_p(t)\alpha(x_s, x_p)/K(x_s)\right]^{\beta}}, \quad (3)$$

$s = 1, \ldots, S$. The sum in the denominator on the right hand side of (3) is the effective population size experienced by phenotype $x_s$. Equation (3) is a discrete time analog of the continuous-time many-species logistic competition model in multidimensional phenotype space that was used in several previous articles[16–18]. In contrast to the continuous time models used previously, in the discrete time model populations can undergo ecological fluctuations and sudden collapses not only after a population itself exceeds the carrying capacity, but also when the cumulative competition from other phenotypes is strong enough (i.e., when the effective population size is above $K$).

For simplicity, and following[15–17], we used the following functions for the competition kernel and the carrying capacity,

$$\alpha(x, y) = \exp\left[-\sum_{i=1}^{d} \frac{(x^i - y^i)^2}{2\sigma_\alpha^2}\right],$$
$$K(x) = K_0 \exp\left[-\frac{\sum_{i=1}^{d} (x^i)^4}{4\sigma_K^4}\right]. \quad (4)$$

Thus, competition is symmetric and strongest between phenotypes that are similar, and the carrying capacity has a unique maximum $K_0$ at 0. We note that in general, using Gaussian forms for both the competition kernel and the carrying capacity can result in structural instabilities[25].

In the second section of Methods, we describe the numerical procedures for simulating the evolutionary process resulting from (3). Typically, simulations start with a single ancestral phenotype, which changes due to mutations and can undergo repeated diversification events due to frequency-dependent competition. In particular, for $\sigma_\alpha < \sigma_K$ in (4), the continuous time analog of the model presented above undergoes adaptive diversification and radiation into a steady state species distribution[13,17,26], see also[27,28]. For example, if we set $\sigma_\alpha = 0.5$ and $\sigma_K = 1$ in (4), then for $\beta = 1$ system (3) is equivalent to the corresponding continuous time system[22], and in a 2-dimensional phenotype space undergoes diversification into a stable community of 16 coexisting phenotypic clusters (species) with approximately constant population sizes. This is illustrated in Fig. 2A and the corresponding video. As long as $\beta = 1$, the observed diversification is independent of $\lambda$ and $K_0$. The main purpose of this paper is to explore the effect of increasing $\beta$ to values >1, which eventually makes the local dynamics (2) unstable. As the exponent $\beta$ is increased, stationary populations lose stability and, similarly to the single-species model shown in Fig. 1, the ecological dynamics of populations in an evolving community become first periodic and then chaotic (see below for specific examples of non-equilibrium dynamics).

For low intrinsic growth rates $\lambda$ this has profound effects on the amount of diversity in the system, as illustrated in Fig. 2 (and accompanying videos of various diversification scenarios corresponding to Fig. 2 can be found here: figshare.com/s/f2d8ecf480fa372319e1). For such $\lambda$-values, increasing $\beta$ in the local dynamics (2) has the effect of eventually inducing pronounced boom-bust population dynamics (cf. Fig. 1). In an evolving community, increasing $\beta$ induces boom-bust dynamics in each of the phenotypes present in the community, with each

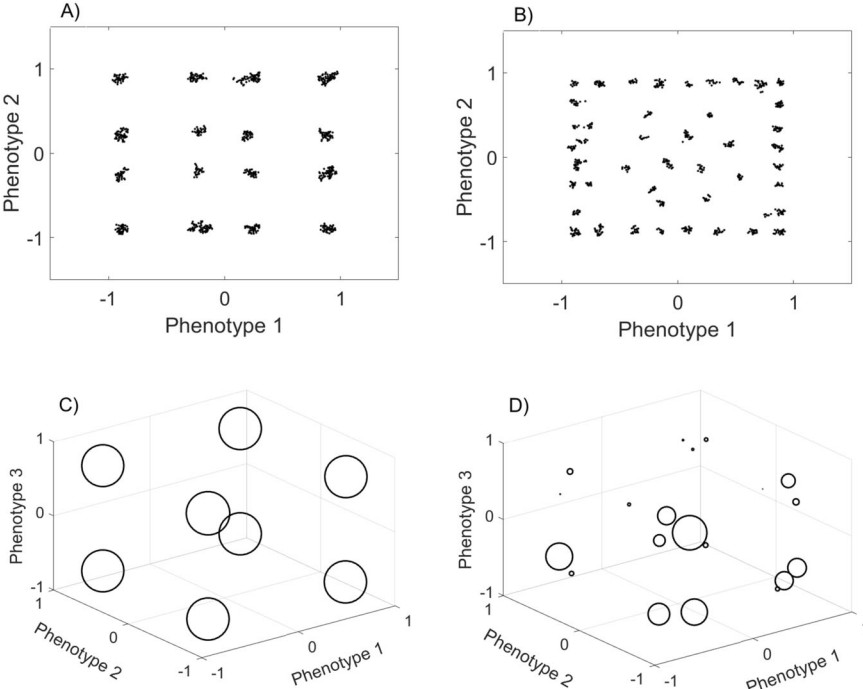

**Fig. 2 Snapshots of cluster distributions.** Distributions are shown for low (left column) and high (right column) $\beta$-values after $10^7$ generations, for 2-dimensional (top row) and 3-dimensional (bottom row) phenotype spaces. **A** 16 species for $\beta = 1$, $\lambda = 1.2$, $\sigma_\alpha = 0.5$; **B** 47 species for $\beta = 45$, $\lambda = 1.2$, $\sigma_\alpha = 0.5$; **C** 8 species for $\beta = 1$, $\lambda = 1.2$, $\sigma_\alpha = 0.75$; **D** 20 species for $\beta = 55$, $\lambda = 1.2$, $\sigma_\alpha = 0.75$. The videos of various diversification scenarios corresponding to Fig. 2 can be found here: figshare.com/s/f2d8ecf480fa372319e1. In the top row panels and corresponding videos (2-dimensional case), all phenotypes present in the evolving community are shown (represented as dots). In the bottom row panels and corresponding videos (3-dimensional case), species resulting from clustering of populations of similar phenotypes using a merging distance of $\Delta x_{species} = 10^{-1}$ (see second section of Methods) are represented by circles whose size is proportional to a species' population size.

phenotypic cluster (species) undergoing multiple generations of (near-)exponential population growth punctuated by deep crashes. These ecological dynamics unfold in such a way that the dynamics of neighboring clusters of phenotypes are desynchronized, i.e., such that crashes and subsequent exponential growth phases occur at different time points. With such desynchronized boom-bust ecological dynamics, evolution can generate a drastic increase in diversity compared to that evolving in ecologically stable communities (Fig. 2). Increased diversity due to boom-bust ecological dynamics typically occurs as long as the ecological conditions for adaptive diversification due to frequency-dependent competition are met (i.e., as long as $\sigma_\alpha < \sigma_K$ in (4)). While the amount of diversity depends on the exact values of $\sigma_\alpha < \sigma_K$, significantly more diverse communities tend to evolve with boom-bust dynamics than with stable equilibrium ecological dynamics.

Figure 3 shows the number of species coexisting at the evolutionary saturation state as a function of the parameter $\beta$ for different values of $\lambda$. The figure illustrates that $\lambda$ has to be small enough for a substantial increase in diversity to be observed for high $\beta$. Indeed, in the local model (2) boom-bust dynamics can only be observed for small $\lambda$, and it is exactly these kinds of population dynamics that allow for increased diversity. For larger $\lambda$, increasing $\beta$ results in more "traditional" forms of chaotic dynamics with irregular, high-frequency oscillations of increasing amplitude. Such local dynamics also lead to chaotic ecological dynamics in populations comprising an evolving community, but they do not generate an increase in the diversity that can evolve and be maintained. In general, as $\beta$ is increased to very high values, the model becomes less relevant biologically:

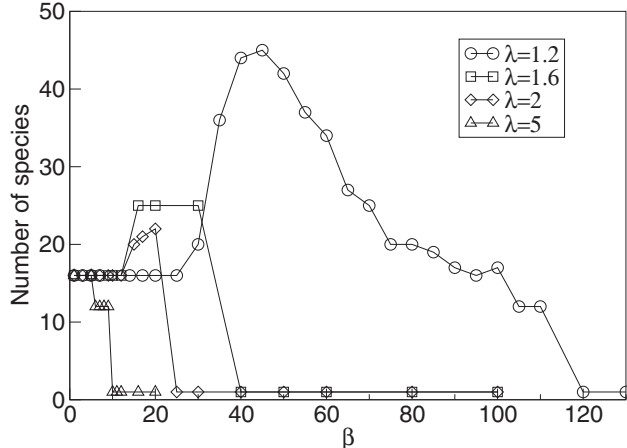

**Fig. 3 Number of coexisting species as a function of the exponent $\beta$.** Circles are for $\lambda = 1.2$, squares for $\lambda = 1.6$, diamonds for $\lambda = 2$, and triangles for $\lambda = 5$. Dimension of phenotype space is $d = 2$, and $\sigma_\alpha = 0.5$. Species are counted after $10^7$ generation and after clustering of populations of similar phenotypes using a merging distance of $\Delta x_{species} = 10^{-1}$ (see second section of Methods).

the population crashes become very severe, which results in extinctions that are frequent enough for diversity not to be able to evolve anymore (see below).

Figure 4 illustrates the desynchronized boom-bust dynamics in an artificial community of 25 species, with each species

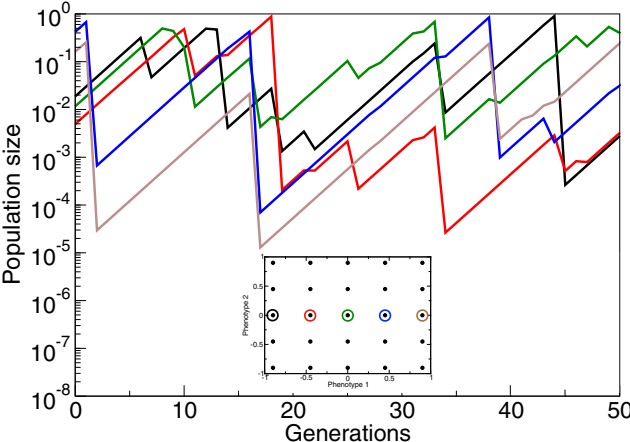

**Fig. 4 Examples of population dynamics of 5 neighboring phenotypes in a community consisting of the 25 phenotypes.** The whole community is indicated by color in the inset. Parameter values are $\beta = 30$, $\lambda = 1.6$, and $\sigma_\alpha = 0.5$, which corresponds to the right-most square with more than 1 species in Fig. 3. Dimension of phenotype space is $d = 2$. With equilibrium population dynamics ($\beta = 1$) the 25 phenotypes cannot coexist. Note that the population dynamics of immediate neighbours are desynchronized.

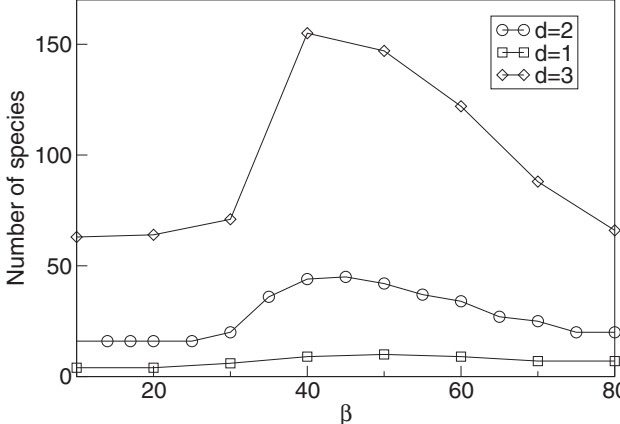

**Fig. 5 The number of species as a function of the exponent $\beta$.** Data are shown for $\lambda = 1.2$, $\sigma_\alpha = 0.5$ and three different dimensions of phenotype space, $d = 1$ (squares), $d = 2$ (circles), and $d = 3$ (diamonds). Species are counted after $10^7$ generation and after merging the populations of similar phenotypes using a merging distance of $\Delta x_{species} = 10^{-1}$ (see description in main text). Note that the level of diversity shown for small $\beta$-values corresponds to the diversity evolving with stable equilibrium ecological dynamics.

represented by a single phenotype, and such that the phenotypes are arranged on a regular grid in phenotype space (see inset in Fig. 4). This community corresponds to the case indicated by the right-most square with more than 1 species in Fig. 3, in which $\beta$ is large enough for the diversity to increase to 25 coexisting species, rather than the 16 that would evolve for lower $\beta$. Figure 4 shows the population dynamics of a subset of 5 phenotypes arranged on a line in the grid, resulting from simulating the ecological dynamics (3) of the whole community of 25 phenotypes. For each phenotype, the dynamics exhibits boom-bust cycles, and neighboring cycles are desynchronized.

It is worth noting that the increased diversity seen for higher $\beta$-values occupies approximately the same phenotypic range as the lower diversity for lower $\beta$-values (Fig. 2). This implies that with higher diversity, the different species are more closely packed in phenotype space, and hence that, generally speaking, conditions of limiting similarity are relaxed in the boom-bust dynamic regime. Because of lower thresholds for limiting similarity, i.e., denser packing, the increase in diversity at evolutionary stationary state due to boom-bust ecological dynamics becomes more pronounced with higher dimensions of phenotype space, as illustrated in Fig. 5.

Relaxed limiting similarity conditions require a decrease in competitive pressure that species in neighboring regions of phenotype space exert on each other. Such a decrease can be achieved if neighboring populations are fluctuating in opposite phases, as shown for an artificial example in Fig. 4. Figure 6 illustrates that in full simulations of the evolutionary process, neighboring species indeed generally exhibit such an anti-correlation for high $\beta$-values. Essentially, the anti-correlation between populations of neighboring species stems from the asynchrony of their boom-bust cycles. In the first section of the Supplementary Material, we show that such desynchronization is expected to emerge spontaneously from an arbitrary small initial difference between populations: in a simple idealized configuration of two competing species with boom-bust dynamics, their population sizes converge to a state of complete anti-synchronization. For smaller values of $\beta$ or larger values of $\lambda$, for which populations do not undergo boom-bust dynamics this anti-correlation is not seen (Fig. 6).

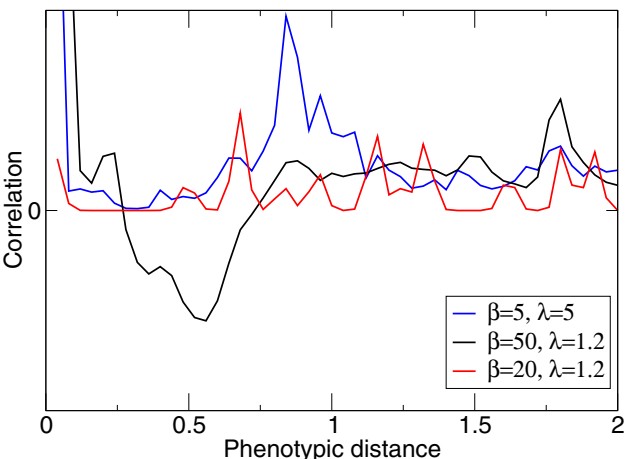

**Fig. 6 Correlation between population sizes of different phenotypes.** The correlation between phenotypes $x$ and $y$, $C_{xy} \equiv \langle (N_x(t) - \langle N_x \rangle)(N_y(t) - \langle N_y \rangle) \rangle$, is shown as a function of phenotypic distance $|x - y|$ for $\beta = 50$, $\lambda = 1.2$ (black line), $\beta = 20$, $\lambda = 1.2$ (red line) and $\beta = 5$, $\lambda = 5$ (blue line). Dimension of phenotype space is $d = 2$, and $\sigma_\alpha = 0.5$. The correlation was calculated by taking into account all possible pairs of phenotypes over $5 \times 10^6$ generations after the steady state level of diversity was reached. Anti-synchronization is only seen for boom-bust dynamics (black) allowing for increased diversity (cf. Fig. 3), but not for higher frequency chaotic dynamics with small (red) or large (blue) amplitudes.

The explanation for higher diversity based on anti-correlated boom-bust cycles of phenotypically close species suggests that to make this mechanism work, these cycles should be of sufficient length. This means that the population crashes should be sufficiently severe (large $\beta$), and the intrinsic growth rate $\lambda$ should be sufficiently small. Essentially, the exponential phase should be long enough for robust desynchronization. This effect cannot be achieved with high intrinsic growth parameters $\lambda$ (Fig. 3): the increase in diversity is noticeably diminished for $\lambda = 1.6$, and is absent for larger $\lambda$. In particular, the type of chaotic population

fluctuations induced by high $\beta$ for $\lambda$-values that are significantly larger than 1 do not lead to increased diversity, because for larger $\lambda$, complex dynamics are not of the boom-bust type.

To confirm that the boom-bust cycles, rather than chaoticity or other features of the population dynamics defined by Eq. (3), are the essential mechanism for the observed increase in diversity, we stripped the model (3) from all other features except its ability to run boom-bust cycles: We assumed that each phenotypic population grows exponentially with an exponent $\lambda$ until the effective density experienced by a given phenotype, i.e., the cumulative competitive effect of all phenotypes, given by the term $\sum_{p=1}^{S} N_p(t)\alpha(x_s, x_p)$ in denominator of (3), becomes greater than the carrying capacity of that phenotype. When that happened, the population of that phenotype was reduced to a small fraction of its population size, simulating a severe crash. The mutation and merging procedures were implemented as in the original model. This modified model shows qualitatively very similar results (not shown) and exhibits significant increases in diversity at a level very similar to the original model, as long as $\lambda$-values are close to 1, so that the exponential phase starting from low densities is long and slow, and as long as the population crashes are severe enough. This confirms that the key for the evolution of higher diversity is the existence of pronounced boom-bust dynamics for all phenotypic populations.

An interesting question concerns the effect of the frequency and size of mutations on diversity. These were assumed to be $\mu = 0.1$ and $\Delta x = 10^{-2}$ for the results presented so far (note that it is really only the product of these two parameters that matters). A reduction in $\mu$ and/or $\Delta x$ slows down evolution in general and diversification in particular. This effect is illustrated in Fig. S.3A, where we show the number of species vs. time for four different mutation frequencies. Smaller mutation rates result in longer times to reach the equilibrium level of diversity. There is, however, another, less direct effect of mutation rate on diversity. For any non-zero extinction threshold and even moderate $\beta$, there is a small but finite probability that all phenotypes of a well-developed cluster, and hence the corresponding species, go extinct during a particularly severe bust. The extinct cluster can eventually get replaced by newly arising mutants, but the time it takes mutations to undergo a sufficient number of phenotypic steps to reach the vacated spot in phenotype space depends on $\mu$ and $\Delta x$. For any given mutation rate and size, these processes may equilibrate at different levels of diversity. In particular, lower extinction thresholds (making extinction less likely) lead to higher levels of diversity at saturation. This is illustrated in Fig. S.3B.

In general, diversity decreases drastically for very high $\beta$-values and eventually the system is reduced to just a single phenotypic cluster. This occurs because with large $\beta$-values, the crashes due to the effective density experienced being higher than the carrying capacity become progressively more severe and can bring all phenotypes comprising one species below the minimum population threshold, thus rendering the species extinct. Even though diversification is still favored by selection, the rate of species extinction for high $\beta$-values is too high for diversity to evolve. The very dynamic regime of this "competition" between extinction and diversification is illustrated in Fig. S.2.

## Discussion

We propose a possible explanation for the emergence and persistence of large amounts of diversity based on competition models for evolving communities with fluctuating population dynamics. When these fluctuations are in the boom-bust regime, in which long periods of exponential growth are followed by deep population crashes, diversity in continuous phenotype spaces evolves well beyond what is expected based on limiting similarity with stationary ecological dynamics. The key mechanism that results in higher diversity is the spontaneous desynchronization of boom-bust cycles between phenotypically similar species, which essentially reduces interspecific competitive impacts and allows for much denser packing of species in niche space.

Population fluctuations have long been considered as a potential mechanism leading to violations of the competitive exclusion principle. For the most part, past studies have either assumed a fixed set of resources[2,9,29], or they have assumed externally imposed fluctuations[10,11]. In such models, the mechanism of ecological fluctuations causing an increase in diversity can be viewed as a form of the temporal storage effect[7,8], which intuitively corresponds to temporal segregation in niche space[10]. In fact, there have also been models showing that population fluctuations can decrease diversity in an evolutionary context[30], but these models appear to allow for jack-of-all-trades mutations on a finite set of resources, which can increase rather than decrease the amount of interspecific competition in the system.

Our models extend previous models for the emergence and maintenance of diversity under stable equilibrium ecological dynamics[13,17,26–28]. They differ from earlier models such as[27,28] in key aspects: they consider evolution in high-dimensional phenotypes that characterize continuously variable and multivariate niche use, and persistent ecological fluctuations are intrinsically generated by overcompensating competition. Desynchronized boom-bust cycles provide a robust mechanism for a substantial increase in the diversity that can evolve and be maintained in such models, an effect that increases with increasing dimension of phenotype space. We note that this latter result is not obvious, as with higher phenotypic dimensions the number of phenotypically similar species (nearest neighbours in phenotype space) increases linearly with the dimension, which may be expected to make desynchronization of neighboring boom-bust cycles more difficult due to denser phenotype packing.

This mechanism of "diversification in time" is similar to those previously reported[10,29]: time acts as additional niche space, and separation along this niche space can alleviate interspecific competition. In the language of[31], boom-bust desynchronization effectively increases the "environmental dimension", which is a determinant of the amount of diversity that can be sustained. Again, this is akin to the temporal storage effect[7,8], although the latter is mostly invoked for externally generated population fluctuations. The longer the boom-bust cycles, the more temporal separation between similar species is possible. If the population crashes in the boom-bust regime become too severe, they produce frequent extinctions, which eventually leads to a net negative effect on diversity.

In our models, higher diversity can only be observed in the presence of pronounced boom-bust cycles, but not with other types of population fluctuations, such as periodic or chaotic dynamics with high-frequency oscillations. From a modeling perspective, it is worth noting that more standard and more widely used discrete-time models, such as the Ricker equation or the discrete-time logistic model, even in their chaotic regimes cannot exhibit the type of chaotic boom-bust dynamics that model (2) exhibits for low intrinsic growth rates $\lambda$ and large (overcompensating) $\beta$. This reiterates old cautionary notes about the judicious use of discrete maps for modeling ecological dynamics[21].

Discrete-time models have proved to be very useful for many different purposes in ecology and evolution at least since Ricker's s famous stock and recruitment paper[24]. However, we note that boom-bust dynamics can also be generated using continuous-time models. To illustrate this, consider a continuous-time analogue of

the modified model introduced at the end of the Results section. This continuous-time model has two phases, representing slow exponential growth and fast exponential decline in continuous time. In the first phase, long and slow exponential growth occurs from low densities for each phenotypic population, while keeping track of the effective density experienced by each phenotype, i.e., of the weighted sum overall phenotypic population sizes, with weights given by the competition kernel (this corresponds to the sum in the denominator of Eq. (3)). Once the effective density of a given phenotype reaches the carrying capacity of that phenotype, there is a very fast exponential decline until the phenotype reaches a small fraction of the population size it had before the decline, which corresponds to a severe population crash. Simulations of this simple boom-bust model in continuous time (results not shown) produce qualitatively identical results: the amount of diversity that emerges and is maintained evolutionarily is much larger than the diversity that would evolve with stable equilibrium dynamics (as e.g., reported using a continuous-time logistic model in[17]). This again underscores the generality of the effects of boom-bust ecological dynamics on diversity.

We speculate that the mechanisms and results reported here are not limited to competition models, but could also be manifest in communities with other ecological interactions, e.g., in communities with crowding effects[32], or in communities containing both predators and prey[33]. Whenever the population dynamics exhibit patterns of rapid growth interspersed by crashes (as may e.g., be expected in many predator-prey systems), temporal desynchronization can occur spontaneously and thus lead to increased diversity. Such effects were shown in[33], who reported that "kill-the-winner" mechanism, in which predation generates crashed in the most abundant consumer species, can generate increased levels of diversity. These mechanism differ from the ones reported here in that they are extrinsic to the crashing consumer species (and it is difficult to compare those system to baseline systems with stable ecological dynamics).

There is some empirical support for the effect of boom-bust cycles on ecosystem diversity. For example, such patterns were observed in carefully staged long-running experiments with several plankton species[6], and the experimental data showed that out-phase oscillations in predator-prey cycles of zooplankton and phytoplankton were important for the maintenance of diversity in this system[34]. Predation from pathogens have also been reported to induce algal boom-bust cycles[35]. Generally, boom-bust cycles appear to be common in many marine ecosystems, which are known to be very diverse. For example, it has been suggested to call echinoderms a "boom-bust" phylum[36], and recent work shows that in polar plankton communities, which constitute an important ecosystem in the global ocean, phytoplankton dynamics are often categorized by "boom-bust" cycles[37].

It is interesting to put our results in the context of observations of "neutral evolution". For example[38], report neutral taxonomic distributions during early metazoan diversification into relatively empty niche space. In our models, such expansions could be classified as the boom stage, and according to our model assumption would then indeed occur essentially unabated and in the absence of competitive effects. The actual selection only occurs during the bust stage with populations of less adapted species crashing earlier and deeper. Such an application of the model (3) would be rather speculative however, as the unrestricted exponential growth phase would have to last for a very long time and would in any case represent a simplified and unrealistic assumption for such scenarios. We also note that[38] consider neutrality based on taxonomic data, not on functional data, whereas our model only considers functional phenotypic data. It has been noted that the distinction between taxonomic and functional data is very important in many microbial ecosystems[39], and in particular that functional data can be decidedly non-random even when taxonomic distributions look random.

Overall, we think that our results provide a useful evolutionary perspective for thinking about diversity in natural ecosystems. Boom-bust population fluctuations are a robust, intuitively appealing and probably under-appreciated potential cause of significantly increased diversity in evolving ecosystems.

## Methods

**Basic model properties.** It is well known that that the basic quantity underlying the dynamic behavior of model (2) is the derivative $dF/dN$ evaluated at the equilibrium $K$:

$$\left.\frac{dF}{dN}\right|_{N=K} = 1 - \frac{\lambda - 1}{\lambda}\beta. \tag{5}$$

For $\lambda > 1$, the population dynamics converges to the steady state $K$ if and only if $|1 - \frac{\lambda-1}{\lambda}\beta| < 1$. Thus, for a given $\lambda$, for small values of the exponent $\beta$ the system exhibits stable equilibrium dynamics, and increasing $\beta$ gives way to a period-doubling route to chaos (Fig. 1). Biologically, increasing $\beta$ can be viewed as reflecting a gradual change from contest to scramble competition[23]. This is reflected by the shape of the per capita number of offspring as a function of population size, given by the right hand side of (2) divided by $N(t)$, and viewed as a function of $N(t)$: for any $\lambda$, and for high $\beta$, the per capita number of offspring is approximately constant until the population size reaches the vicinity of $K$, but as $N$ increases above $K$ the number of offspring falls rapidly to very low values, essentially generating a population crash as soon as the population size is above $K$.

**Procedures for evolutionary simulations.** To simulate the evolutionary process, we set the scaling parameters $K_0$ and $\sigma_K$ to $K_0 = 1$ and $\sigma_K = 1$, and we start with a number $S$ of phenotypes (typically, $S = 1$, and the phenotype is randomly chosen in the vicinity of the maximum of the carrying capacity). We then simulate the ecological dynamics in discrete time, using (3) for each of the phenotypes. In each generation, a new phenotype is generated with a probability $\mu$ (typically $\mu = 0.1$). The new phenotype is a mutant of one of the existing phenotypes. Of those, a parental phenotype is chosen with a probability proportional to its population size, and the offspring phenotype is chosen randomly from a Gaussian distribution with the average centered at the parental phenotype and a small standard deviation $\Delta x$ (typically $\Delta x = 10^{-2}$).

After addition of the new phenotype, the community now comprises $S + 1$ phenotypes, and the process is repeated for many generations. What one wants to know from this process is how the distribution of phenotypes changes over time. To keep the number of phenotypes from increasing to very large numbers that would render the simulation computationally impossible, we periodically merge phenotypes that are very close together. Specifically, once every $t_{merge}$ generations (typically $t_{merge} = 1000$), phenotypes that are within a distance $\Delta x$ of each other are merged (preserving their phenotypic center of mass) and their population sizes added. In addition, every generation all clusters with populations densities below a threshold (typically $= 10^{-12}$) are declared extinct and removed from the system. Together, these procedures preserve the phenotypic variance necessary for evolution, but prevent undesirable computational complexity.

To define and count the number of phenotypically distinct species in the community at any given point in time, with each species possibly consisting of a number of similar phenotypes, the phenotypes in the community are clustered with a larger distance $\Delta x_{species}$ (typically $\Delta x_{species} = 10^{-1}$). This phenotypic distance is still significantly smaller than the typical scales of ecological interactions as long as $\sigma_\alpha$ and $\sigma_K$ in (4) are of order 1. This implies that the phenotypes within a designated species experience very similar competitive interactions and generally follow the same population dynamics. Note that species designation is only used to gather statistical data from simulations, but not in the actual computational steps of the simulations.

**Reporting summary.** Further information on research design is available in the Nature Research Reporting Summary linked to this article.

## Data availability

All data used in this study was generated by computer simulations. The data that support the findings of this study are available from the authors upon reasonable request. The movie files referred to in the context of Fig. 2 can be found at videos of various diversification scenarios corresponding to Fig. 2 can be found here: figshare.com/s/f2d8ecf480fa372319e1.

## Code availability

The code for the computer simulations performed for this study was written in Fortran and can be found here: https://github.com/jaros007/

Fortran_code_Boom_bust_population_dynamics_increase_diversity_in_evolving_competitive_communities, and as described in ([Code depository] The code depository is https://doi.org/10.5072/zenodo.747095.).

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

## Acknowledgements
M.D. was supported by NSERC Discovery Grant 219930. E.C.J. was supported funded by the National Agency for Research and Development (ANID) Scholarship 21190785. IY.I. acknowledges support from FONDECYT project 1200708.

## Author contributions
M.D. provided the basic idea, performed simulations, and wrote the paper. E.C.J. performed simulations and data analysis. IY.I. made conceptual contributions, performed simulations and analysis, and wrote the paper.

## Competing interests
The authors declare no competing interests.
