## [Peer Review File · Communications Biology]

Reviewers' comments:

Reviewer #1 (Remarks to the Author):

This paper introduces a discrete-time, multispecies competition model to explore a novel mechanism by which species can coexist in evolving communities. The mechanism involves a phase of effectively uncoupled and uncorrelated exponential species growth (boom phase) and a sudden population crash (bust phase). This boom-bust population dynamics would increase diversity in evolving competitive communities.

Authors' ideas are speculative but powerful. Some empirical evidence for this kind of dynamics in nature exist for certain animal Phyla, such as Echinoderma.

I will make some comments and remarks the authors may want to consider to improve their ms.

1. There is a general worry about the use of discrete-time models in population ecology. For instance, we all know that time-continuous population logistic growth is very simple, while the (time-discrete) logistic map give rise to a variety of dynamic behaviors (including chaos). Although this is briefly commented in some parts of the ms, I believe it need a more careful consideration.

2. The model presented gives rise to boom-bust dynamics only in a region of model parameter space. I have missed a careful characterization of the parameter space, at least, along two relevant axes. No doubt. This would make this contribution stronger

3. Phenotypic mutations occur at a certain pace, but, probably, at much lower pace than ecological dynamics. This eco-evolutionary model makes these two different time scales overlap. A question arises. What is the importance this time scale separation for the appearance of boom-bust dynamics in certain regions of the parameter space? In other words, are authors' results robust to phenotypic mutations occurring at a much lower pace in relation to ecological dynamics?

4. This is a concern in relation to the decay of diversity as beta increases (Fig 2 and 5). Could this be regarded as an artifact of the selected extinction threshold? If this is right, this should be admitted more openly because no ecological interpretation is possible from a model artifact.

5. I believe a better connection to previous literature is possible. Authors' discrete model should be able to fully recover van Nes and Scheffer (2008) results on a one dimension trait space, which seems to be the case, but it is not acknowledged. Furthermore, the formation of species stable clusters as phenotypic evolution occurs has been understood in terms of a diffusion-like instability over trait space which is sensitive to the functional form of the competition kernel (Hernandez-Garcia et al, 2009). Could the authors relate cluster formation to this type of instability? Relevant missed references are:

Scheffer, M., & van Nes, E. H. (2006). Self-organized similarity, the evolutionary emergence of groups of similar species. *Proceedings of the National Academy of Sciences of the United States of America*, 103(16), 6230–6235. <https://doi.org/10.1073/pnas.0508024103>

Hernández-García, E., López, C., Pigolotti, S., & Andersen, K. H. (2009). Species competition: coexistence, exclusion and clustering. *Philosophical Transactions. Series A, Mathematical, Physical, and Engineering Sciences*, 367(1901), 3183–3195. <https://doi.org/10.1098/rsta.2009.0086>

6. If biological diversity has been generated and maintained by boom-bust dynamics, an interesting view of the eco-evolutionary process emerges. Biological diversity would unfold through effectively independent exponential growth of species populations (most of the time), punctuated by (sudden) extinctions. If this picture is true, it would explain why certain effective descriptions of community dynamics work although they are based on over-simplifying assumptions (plainly wrong in nature), such as species-independence or species equivalence (as assumed in neutral models). For instance, "early metazoan diversification may not have been driven by systematic adaptations to the local environment, but instead may have resulted from stochastic demographic differences", so the structure of certain paleocommunities is determined by the dominance of

neutral processes (Mitchell et al, 2019). It would be nice to see some comments about this in the discussion section.

Mitchell, E. G., Harris, S., Kenchington, C. G., Vixseboxse, P., Roberts, L., Clark, C., Dennis, A., Liu, A. G., & Wilby, P. R. (2019). The importance of neutral over niche processes in structuring Ediacaran early animal communities. *Ecology Letters*, 22(12), 2028–2038.
<https://doi.org/10.1111/ele.13383>

Finally, just a typo: In line 235, I guess you meant "beta-values".

Reviewer #2 (Remarks to the Author):

Summary

The manuscript investigates the problem of paradox of plankton, which is a long-term problem in ecology about how high diversity maintains in ecosystems, by studying a model of difference equations of competing population densities with evolving multi-dimensional phenotypes and phenotype-dependent carrying capacity. From simulation results, the authors claim that population fluctuation from the difference equations, under the condition of strong competition and high dimensional phenotype space, can lead to asynchronous "boom-bust" dynamics of many coexistent species, therefore maintaining high diversity.

The authors present an interesting work by starting with a simple model. Diversity that can be sustained by strong competition seems interesting and counter-intuitive. In principle this work could provide some hints to other theoretical studies with more realistic and complicated cases that contain trophic levels such as predator-prey systems or food webs. However, I have reservations about the manuscript. The main issue that I found is the physical meaning and implication of the results, as well as the further applicability of the model, which are not well discussed in the manuscript. Another issue I found is the lack of literature search, as biodiversity and paradox of plankton have been widely studied over the decades, so it is not clear the new contribution from this manuscript. In this present form of the manuscript, I could not recommend publication. I encourage the authors to improve the manuscript and clarify the description as suggested below, and therefore the manuscript could be worth reconsidering publication.

1. It is not clear how the authors obtain the model in Eq (1) and the therefore the following Eq (3). As the authors state, the difference equation can be exactly derived for $\beta=1$ by solving the logistic growth equation. However, how the solution is directly extended to " $\beta > 1$ " is not clear to me. Eq (1) seems unlikely the solution of the growth equation dN/dt with a generalized term $(N/K)^\beta$ which could be interpreted as many-body interactions or the crowding effect. Therefore Eq (1) looks more of an ad hoc artificial model, and I found it confusing when the authors state that Eq (1) is "the basic ecological model". Although the authors claim that Eq (1) can fit a wide range of data as shown in Ref (20), the data in the reference seem to be able to be fitted with different types of models. I suggest the authors to specify whether and how Eq (1) and (3) are derived, and before Eq (1) state why the authors choose such an update function or why it is commonly adopted by the community.

2. Following Comment#1, I find it is hard to understand the "competition" defined by the authors for $\beta > 1$. I would regard the magnitude of the competition terms as the strength of competition. For example, if β is larger but the coefficient in front of $(N/K)^\beta$ is small, then I would consider it is still a case under weak competition. In Eq. (1) the competition effect from β is coupled with strength of the intrinsic growth rate λ , and therefore it is confusing to discuss the effect of λ . In fact, to reduce the number of parameters, the intrinsic growth rate is usually eliminated by rescaling the time in the growth equation dN/dt , even with a generalized term $(N/K)^\beta$ in dN/dt . In this sense, the ad hoc equation in Eq (1) with more and dependent parameters seems not only more complicated but also ambiguous for biological interpretations. For example, in Fig 3 could the lower diversity with large values of λ be an effect of strong competition actually? I suggest the authors to define the competition more clearly and discuss the possible intertwined effect of λ and β .

3. Another issue I found confusing is that there is not enough literature as it should be. Biodiversity and paradox of plankton have been a big topic in ecology and there has been a huge amount of proposed models and mechanisms, including the effect of demographic fluctuations, evolutionary traits, spatial inhomogeneity etc. For example, a paper by Gavina et al, Sci Rep. 8(1), 1198 (2018) considering the higher order of interspecies competition shows that the coexistence can be sustained by weak interspecies competition from the crowding effect, which seems to be opposite to the authors' claim. Also the phenomenon similar to the asynchronous "boom-bust" dynamics due to evolution and stochasticity is studied in the paper by Xue and Goldenfeld, Phys. Rev. Lett. 119, 268101 (2017). I suggest the authors to include more literature in the introduction and indicate how the authors' model is different from them or justify why the authors choose the feature in their model.

4. Below Eq (2), it is not clear to me why λ has to be larger than 1 to avoid extinction. I suggest that the authors explain more how this is obtained.

5. Following Comment#4, it is not clear to me why the population dynamics converges to a steady state if and only if " $|1 - ((\lambda - 1)/\lambda)^\beta| < 1$ ". Do you mean " $|1 - ((\lambda - 1)/\lambda)^\beta| < 0$ " instead? I suggest that the authors briefly explain this or list the reference for it.

6. Higher diversity in higher dimension seems to be straightforward to me, with or without competition, because the phenotype space is larger. I suggest the authors to illustrate more about the purpose of studying higher dimension.

7. In the authors' model the growth rate is independent of phenotype and there is no trade-off. It seems unnatural and I suggest the authors to justify this assumption.

End of comments

Reply to reviewer comments on "Boom-bust population dynamics can increase diversity in evolving competitive communities", submitted to *Communications Biology*

November 29, 2020

Reviewer comments in italics

Replies in regular font

Changes to the manuscript in blue

Reviewer 1:

1. There is a general worry about the use of discrete-time models in population ecology. For instance, we all know that time-continuous population logistic growth is very simple, while the (time-discrete) logistic map give rise to a variety of dynamic behaviours (including chaos). Although this is briefly commented in some parts of the ms, I believe it need a more careful consideration.

Reply: We would like to point out that discrete time models have been used in population biology for roughly 100 years, first starting in population genetics, and in ecology at least since Ricker's famous 1954 paper on fisheries, and then very prominently since the 70's when it was realized that in contrast to continuous-time models, discrete maps can easily capture many different types of ecological dynamics. Discrete-time models have been used in thousands of publications, ranging from purely theoretical to very applied settings, and many publications have pointed out their potential usefulness as realistic ecological modes (including R. May's well-cited review article published in Nature in 1976, which includes multiple references on the population dynamic roots of many simple discrete maps; in fact, in a variety of ecological scenarios, such as those of seasonally and synchronously reproducing animals, insects, plants, and fungi, the discrete generation models provide more realistic descriptions than continuous-time models; see also our reply to the first comment by reviewer 2). Referring to a "general worry" about such models is in our view somewhat disingenuous, as one might say the same thing about models in general.

That said, for our work the particular choice of discrete-time model is of no importance as long as it can exhibit boom-bust dynamics. Rather, what we want show in our paper is that if ecological dynamics are of a certain type, specifically of boom-bust type consisting of long, slow exponential phases followed by a population crash, then much more diversity can be maintained than with equilibrium ecological dynamics. Any model that can exhibit this

type of boom-bust dynamics could be used for our arguments. It so happens that simpler (2-parameter) discrete time models cannot exhibit such behaviour, but the model we used in the paper can, specifically for low λ -values and high β -values. For our argument, the region in parameter space in terms of model parameters for which the model exhibits these dynamics is not relevant, as we are not trying to assess the "likelihood" of boom-bust dynamics. We simply make the argument that if (and only if) the ecological dynamics are of boom-bust type, then more diversity is maintained. To repeat, any model that exhibits this type of dynamics would serve for our purposes, regardless of what other dynamics the model might exhibit.

In fact, it is easy to come up with a continuous-time model that would yield the same results. Such a model would have slow exponential growth (in continuous time) from low densities, while keeping track of the "effective density" that each population experiences, i.e., of the sum in the denominator of eq. (4) in the manuscript. As soon as the effective density experienced by a particular phenotype exceeds the carrying capacity of that phenotype, the model would then have a very fast negative exponential period (again in continuous time), simulating a population crash to a very small population size, after which the slow exponential growth starts anew. We have checked using our simulations that this type of dynamics has the same effect on diversity (as expected, because this is exactly the type of dynamics that unfolds in our discrete-time model). Of course this model is also artificial, in that it requires keeping track of the effective density, but the thought experiment shows that the details of the model formulation do not matter as long as the model exhibits pronounced boom-bust dynamics. Because we are not making any claims about the likelihood of such dynamics, we think it would be distracting from the main message to compare parameter ranges for which boom-bust does or does not occur.

Changes made:

- We added the following paragraph after the 3rd paragraph in the model section:

We note that there are in principle many different models that can exhibit boom-bust dynamics (including models set in continuous time, see Discussion section). We chose model (1) as a generic model with boom-bust dynamics for certain parameter regions, viz. for λ -values close to 1 and large enough β . Rather than being interested in the likelihood of a particular model exhibiting boom-bust dynamics, we are interested in the consequences of such dynamics for the evolution of diversity. Therefore, while pointing out the contrast to the consequences of other types of ecological dynamics, such as cyclic or "regular" chaotic dynamics, delineating the different regions in parameter space generating the different types of dynamics is not relevant for our purposes.

- We added the following new paragraph in the Results section section (3rd last):

To confirm that the boom-bust cycles, rather than chaoticity or other features of the population dynamics defined by Eq. (4), are the essential mechanism for the observed increase in diversity, we stripped the model (4) from all other features except its ability to run boom-bust cycles: We assumed that each phenotypic population grows exponentially with an exponent λ until the effective density experienced by a given phenotype x_s , i.e., the cumulative competitive effect of all phenotypes, given by the term $\sum_{p=1}^S N_p(t)\alpha(x_s, x_p)$ in denominator of (4), becomes greater than the carrying capacity of that phenotype. When that happened, the population of that phenotype was reduced to a small fraction of its population size, simulating a severe crash. The mutation and merging procedures were implemented as in the original model. This modified model shows qualitatively very similar results (not shown) and exhibits significant increases in diversity at a level very similar to the original model, as long as λ -values are close to 1, so that the exponential phase starting from low densities is long and slow, and as long as the population crashes are severe enough. This confirms that the key for the evolution of higher diversity is the existence of pronounced boom-bust dynamics for all phenotypic populations.

- We added the following paragraph after the 4th paragraph in the Discussion section:

Discrete-time models have proved to be very useful for many different purposes in ecology and evolution at least since Ricker's famous stock and recruitment paper (Ricker 1954). However, we note that boom-bust dynamics can also be generated using continuous-time models. To illustrate this, consider a continuous-time analogue of the modified model introduced at the end of the Results section. This continuous-time model has two phases, representing slow exponential growth and fast exponential decline in continuous time. In the first phase, long and slow exponential growth occurs from low densities for each phenotypic population, while keeping track of the effective density experienced by each phenotype, i.e., of the weighted sum over all phenotypic population sizes, with weights given by the competition kernel (this corresponds to the sum in the denominator of eq. (4)). Once the effective density of a given phenotype reaches the carrying capacity of that phenotype, there is a very fast exponential decline until the phenotype reaches a small fraction of the population size it had before the decline, which corresponds to a severe population crash. Simulations of this simple boom-bust model in continuous time (results not shown) produce qualitatively identical results: the amount of diversity that emerges and is maintained evolutionarily is much larger than the diversity that would evolve with stable equilibrium dynamics (as e.g. reported using a continuous-time logistic model in (Doebeli and Ispolatov 2017)). This again underscores the generality of the effects of boom-bust ecological dynamics on diversity.

2. *The model presented gives rise to boom-bust dynamics only in a region of model parameter space. I have missed a careful characterization of the parameter space, at least, along two relevant axes. No doubt. This would make this contribution stronger.*

Reply: As mentioned, the primary goal of this work is to show that a higher than usual level of diversity could emerge in evolving systems with boom-bust dynamics. The model defined in Eqs. (1-2) serves as an example to explore this phenomenon. As explained above, one can come up with many other models (perhaps not as minimalistic mathematically as (1,2)) that could generate boom-bust dynamics, even in continuous time, as discussed now in the revised manuscript (see above reply to point 1).

We have also explained at length (3rd and 4th paragraph of Model and simulation methods section, 2nd, 3rd and 6th paragraph of Results section) that what is needed for boom-bust dynamics are long (and slow) exponential growth phases interspersed by population crashes when the effective density grows above the carrying capacity. In the discrete time model (1-2) this requires λ -values not much above 1 (so that exponential growth from low densities is not too fast), as well as high β -values, which ensure severe population crashes. Increased diversity only results for such parameters, as described in our Figure 3, as well as in the main text (2nd, 3rd and 6th paragraph of Results section).

In our opinion, yet another figure characterizing the model in terms of parameters would distract from the main message of the manuscript. Again, we are not interested in assessing the likelihood of boom-bust dynamics in a particular model. Besides, the range of parameters that leads to the required boom-bust dynamics can only be described qualitatively, and for example depends on the extinction threshold (see our reply to point 4), so that such a phase diagram could be defined only imprecisely.

Changes made:

- See changes made according to the reply to the previous point.

3. *Phenotypic mutations occur at a certain pace, but, probably, at much lower pace than ecological dynamics. This eco-evolutionary model makes these two different time scales overlap. A question arises. What is the importance this time scale separation for the appearance of boom-bust dynamics in certain regions of the parameter space? In other words, are authors' results robust to phenotypic mutations occurring at a much lower pace in relation to ecological dynamics?*

Reply: This is a very good point, and we thank the Reviewer for bringing it up. Indeed, a reduced mutation frequency directly slows down evolution in general and diversification in particular. We now illustrate this effect in a new figure in the Supplementary material. Smaller mutation rates results in longer times to reach the equilibrium level of diversity.

There is, however, another, less direct effect of mutation rate on diversity: for any sufficiently small extinction threshold and even moderate β , there is a small but finite probability that a well-developed cluster goes extinct during a particularly severe bust. The extinct cluster eventually gets replaced by newly arising mutants, yet the time it takes mutants to undergo sufficient number of phenotypic steps to reach the vacated spot in phenotype space increases for a smaller rate and phenotypic effect (size) of mutations. Thus, for a given mutation rate and size, higher extinction thresholds tend to result in slightly lower diversity (which are nevertheless still significantly elevated compared to levels without boom-bust dynamics).

Finally, we would like to point out that even the mutation rate (and size) assumed for the results in the main text is quite small: we assume that on average there is one new mutant every 10 generations in the entire evolving community, and if a mutant occurs its typical size is less than 1% of the parental phenotype. Given that our phenotypes should be thought of as generalized continuous traits such as body size or metabolic rate, this seems like a rather conservative assumption about the appearance of new types in an evolving community. (For example, for microbes even as much as one mutation per generation per individual seems realistic.)

Changes made:

- We added the following paragraph to the Results section (new 2nd to last paragraph):

An interesting question concerns the effect of the frequency and size of mutations on diversity. These were assumed to be $\mu = 0.1$ and $\Delta x = 10^{-2}$ for the results presented so far (note that it is really only the product of these two parameters that matters). A reduction in μ and/or Δx slows down evolution in general and diversification in particular. This effect is illustrated in Figure S.3A, where we show the number of species vs. time for 4 different mutation frequencies. Smaller mutation rates result in longer times to reach the equilibrium level of diversity. There is, however, another, less direct effect of mutation rate on diversity. For any non-zero extinction threshold and even moderate β , there is a small but finite probability that all phenotypes of a well-developed cluster, and hence the corresponding species, go extinct during a particularly severe bust. The extinct cluster can eventually get replaced by newly arising mutants, but the time it takes mutations to undergo a sufficient number of phenotypic steps to reach the vacated spot in phenotype space depends on μ and Δx . For any given mutation rate and size, these processes may equilibrate at different levels of diversity. In particular, lower extinction thresholds (making extinction less likely) lead to higher levels of diversity at saturation. This is illustrated in Figure S.3B.

Figure S.3: A: Number of species in the community as a function of time for different mutation rates μ . Species were counted using the clustering algorithm described in the Model and simulation methods section. Other parameter values were as for Fig. 2B in the main text. B: Number of species in the community as a function of time for two different extinction thresholds and for a mutation rate $\mu = 0.005$. Other parameter values were as for Fig. 2B in the main text. For the two scenarios shown in B, videos of the evolutionary process can be found at figshare.com/s/f2d8ecf480fa372319e1.

- We added a new section to the Supplementary Material:

Effect of mutation rate on the evolution diversity

Figure S.3A shows the number of species as a function of time (in generations) for different mutation rates (other parameters are as for Figure 2B in the main text). The lower the mutation rate, the longer it takes for diversity to reach saturation levels. Figure S.3B illustrates that the extinction threshold has an effect on the saturation levels of diversity: lower thresholds (fewer extinctions) lead to elevated levels of diversity. For Figure S.3B, the mutation rate was $\mu = 0.005$, so that one mutation of typical size less than 1% of the parental phenotype occurs every 200 generations in the entire evolving community.

4. *This is a concern in relation to the decay of diversity as beta increases (Fig 2 and 5). Could this be regarded as an artifact of the selected extinction threshold? If this is right, this should be admitted more openly because no ecological interpretation is possible from a model artifact.*

Reply: Yes, this is correct. In the revised manuscript we now acknowledge the diminished ecological irrelevance of the large- β regime. Again, our main goal in this work was not to study all dynamical regimes of map (1-2), but to investigate an increase in diversity under boom-bust dynamics. Thus we do not further elaborate on this artifact.

Changes made:

- Diminished ecological relevance is now mentioned at the end of the 3rd paragraph of the Results section, and the wording in the last paragraph of the Results section has been adjusted.

5. *I believe a better connection to previous literature is possible. Authors' discrete model should be able to fully recover van Nes and Scheffer (2008) results on a one dimension trait space, which seems to be the case, but it is not acknowledged. Furthermore, the formation of species stable clusters as phenotypic evolution occurs has been understood in terms of a diffusion-like instability over trait space which is sensitive to the functional form of the competition kernel (Hernandez-Garcia et al, 2009). Could the authors relate cluster formation to this type of instability? Relevant missed references are:*

Scheffer, M., & van Nes, E. H. (2006). Self-organized similarity, the evolutionary emergence of groups of similar species. Proceedings of the National Academy of Sciences of the United States of America, 103(16), 6230

Hernández-García, E., López, C., Pigolotti, S., & Andersen, K. H. (2009). Species competition: coexistence, exclusion and clustering. Philosophical Transactions. Series A, Mathematical, Physical, and Engineering Sciences, 367(1901), 3183.

Reply: These papers are indeed relevant, but it is important to note that they describe “baseline” levels of diversity evolving under stable equilibrium ecological dynamics, relative to which we measure the 2-3 fold increase in the number of species in the boom-bust (non-steady state) regime, which is the main result of our work. Also, it is important to point out that the species designation e.g. in Scheffer et al (2006) is problematic: whereas in our setting, a species consists of a cluster of populations with very similar phenotypes, Scheffer et al (2006) call each phenotypic population in such a cluster a species, so that there is coexistence of many “species” with very similar phenotypes. Such a species designation does not make sense to us biologically.

Changes made:

- We added the two references to the first sentence following sentence in the Results section.
- We changed the first two sentences of the 3rd paragraph of the Discussion section.

6. *If biological diversity has been generated and maintained by boom-bust dynamics, an interesting view of the eco-evolutionary process emerges. Biological diversity would unfold through effectively independent exponential growth of species populations (most of the time), punctuated by (sudden) extinctions. If this picture is true, it would explain why certain effective descriptions of community dynamics work although they are based on over-simplifying assumptions (plainly wrong in nature), such as species-independence or species equivalence (as assumed in neutral models). For instance, "early metazoan diversification may not have been driven by systematic adaptations to the local environment, but instead may have resulted from stochastic demographic differences", so the structure of certain paleocommunities is determined by the dominance of neutral processes (Mitchell et al, 2019). It would be nice to see some comments about this in the discussion section. Mitchell, E. G., Harris, S., Kenchington, C. G., Vixseboxse, P., Roberts, L., Clark, C., Dennis, A., Liu, A. G., & Wilby, P. R. (2019). The importance of neutral over niche processes in structuring Ediacaran early animal communities. *Ecology Letters*, 22(12), 2028.*

Reply: This is another very good point. We thank the Reviewer for bringing it to our attention. The scenario considered in the suggested reference, related to an early colonization of relatively empty niches, would be classified as the "boom" phase in the framework of our models. Indeed, in these models such a rapid expansion occurs essentially unabated and in the absence of any competitive effects (as growth is exponential and the species remain well below the environmental carrying capacity. The actual selection occurs during the bust stage with populations of less well adapted phenotypes crashing deeper. However, concluding that neutrality is an important factor based on our models may be a bit too speculative: the unrestricted (neutral) exponential growth for all species rate clearly represents a model assumption due to the simplicity of the model, rather than to realistic features common in such ecological scenarios. It is also important to note note that the paper by Mitchell et al considers neutrality based on taxonomic data, not on functional data. It has been noted that the distinction between taxonomic and functional data is very important in many microbial ecosystems (Louca, Parfrey and Doebeli, Science 2016), and in particular that functional data can be decidedly non-random even when taxonomic distributions look random.

Changes made:

- We added a new paragraph discussing this issue at the end of the Discussion section:

It is interesting to put our results in the context of observations of “neutral evolution”. For example, Mitchell et al (2019) report neutral taxonomic distributions early meta-zoan diversification into relatively empty niche space. In our models, such expansions could be classified as the boom stage, and according to our model assumption would then indeed occur essentially unabated and in the absence of competitive effects. The actual selection only occurs during the bust stage with populations of less adapted species crashing earlier and deeper. Such an application of model (4) would be rather speculative however, as the unrestricted exponential growth phase would have to last for a very long time and would in any case represent a simplified and unrealistic assumption for such scenarios. We also note that Mitchell et al (2019) consider neutrality based on taxonomic data, not on functional data, whereas our model only considers functional phenotypic data. It has been noted that the distinction between taxonomic and functional data is very important in many microbial ecosystems (Louca et al, Science 2016), and in particular that functional data can be decidedly non-random even when taxonomic distributions look random.

Finally, just a typo: In line 235, I guess you meant "beta-values".

Reply: Corrected.

Reviewer 2:

1. It is not clear how the authors obtain the model in Eq (1) and therefore the following Eq (3). As the authors state, the difference equation can be exactly derived for $\beta = 1$ by solving the logistic growth equation. However, how the solution is directly extended to $\beta > 1$ is not clear to me. Eq (1) seems unlikely the solution of the growth equation dN/dt with a generalized term $(N/K)^\beta$ which could be interpreted as many-body interactions or the crowding effect. Therefore Eq (1) looks more of an ad hoc artificial model, and I found it confusing when the authors state that Eq (1) is "the basic ecological model". Although the authors claim that Eq (1) can fit a wide range of data as shown in Ref (20), the data in the reference seem to be able to be fitted with different types of models. I suggest the authors to specify whether and how Eq (1) and (3) are derived, and before Eq (1) state why the authors choose such an update function or why it is commonly adopted by the community.

Reply: Perhaps the first sentence of our Discussion section was misleading: our paper is not primarily about investigating the paradox of the plankton. It is about showing that a

certain type of ecological dynamics, boom-bust dynamics, can facilitate the evolution and maintenance of large amount of diversity. Thus, the paper presents a conceptual perspective on the problem of diversity, rather than a solution to a specific paradox. Naturally, since many plankton species may arguably exhibit boom-bust dynamics, we speculate about the possible applicability of our conceptual ideas to the paradox of the plankton, but we certainly don't view our paper as solving this paradox, or even as specifically intended to investigate it.

That said, we also want to clarify that Eqs. (1-2) is the basic ecological model **in our paper**, not in general (we agree with the reviewer of course that there is no general "basic" model). We chose Eq. (1) as the basic model for our purposes because it can exhibit boom-bust cycles. It is basic in the sense that it serves as the basis for the general multi-species model given by Eq. (4).

Discrete-time equations, which include Eqs. (1,2) and (4), have been extensively used to model population dynamics since the 1970's (and even earlier; see also our response to point 1 of reviewer 1). In a variety of ecological scenarios, such as those of seasonally and synchronously reproducing animals, insects, plants, and fungi, the discrete generation models provide more realistic descriptions than continuous-time models.

It is important to note that for general parameters, the discrete-time models given by equations Eqs. (1,2) and (4) cannot be mathematically derived from simple underlying continuous-time models! This is because simple 1-dimensional continuous-time models can only give rise to equilibrium population dynamics. Therefore, any model that does not exhibit stable (monotonic) equilibrium dynamics cannot be derived from such continuous-time models. In fact, this is one of the important points about discrete-time models: they can give rise to much richer dynamical behaviour than their corresponding continuous-time analogs, which is why they have been used very extensively to model non-equilibrium ecological dynamics. On the other hand, it is true that every simple continuous-time model can be derived from a corresponding discrete-time model (e.g. the logistic continuous-time model can be derived from Eq. (2) by setting $\beta = 1$ and taking time derivatives. Thus, if anything, in a mathematical sense discrete-time models are more fundamental than continuous-time models.

The fact that discrete-time models can in general not be derived from continuous-time models does not make them any less useful or less fundamental: after all, any continuous-time model is in the end also derived in an ad hoc manner. For example, the logistic continuous-time model is obtained by simply assuming that the growth rate declines linearly with density. There is, in our opinion, nothing that makes the continuous-time models somehow more basic or relevant than discrete-time models. As we explain in our response to reviewer 1, it is in fact possible to generate a continuous-time model that exhibits boom-bust dynamics, based on exponential growth, but with the assumption that the sign and magnitude of exponential growth reverses when the effective density experienced by a given phenotype reaches the carrying capacity.

Changes made:

- We changed the first sentence to the Discussion section to:

We propose a possible explanation for the emergence and persistence of large amounts of diversity based on competition models for evolving communities with fluctuating population dynamics.

- We added a paragraph to the Discussion section about continuous-time boom-bust dynamics (see our reply to point 1 by Reviewer 1).

2. Following Comment 1, I find it is hard to understand the "competition" defined by the authors for $\beta > 1$. I would regard the magnitude of the competition terms as the strength of competition. For example, if β is larger but the coefficient in front of $(N/K)^\beta$ is small, then I would consider it is still a case under weak competition. In Eq. (1) the competition effect from β is coupled with strength of the intrinsic growth rate λ , and therefore it is confusing to discuss the effect of λ . In fact, to reduce the number of parameters, the intrinsic growth rate is usually eliminated by rescaling the time in the growth equation dN/dt , even with a generalized term $(N/K)^\beta$ in dN/dt . In this sense, the ad hoc equation in Eq (1) with more and dependent parameters seems not only more complicated but also ambiguous for biological interpretations. For example, in Fig 3 could the lower diversity with large values of λ be an effect of strong competition actually? I suggest the authors to define the competition more clearly and discuss the possible intertwined effect of λ and β .

Reply: We use the model (1-2) to produce boom-bust dynamics, which is an empirically well-known population dynamic pattern, and to show that other forms of dynamics exhibited by this model for other values of parameters, such as chaotic dynamics with short growth intervals, do not lead to enhanced diversity. As is explained in the manuscript, the two parameters λ and β have the following qualitative meaning. In accordance with any other discrete-time model, λ describes population growth in the absence of competition, i.e., when densities are low. In that case, the denominator of Eq. (2) is approximately 1, and hence the population dynamics becomes $N(t + 1) = \lambda N(t)$, i.e., exponential growth, which occurs as long as the denominator of Eq. (2) in the main text is approximately 1, i.e. as long as, in the single species model (2), the population density is small. On the other hand, when the population density increases above the carrying capacity, so that the expression $N(t)/K$ in the denominator of (2) is > 1 , the parameter β becomes important: if β is large, the expression $(N(t)/K)^\beta$ becomes very large as soon as $N(t)/K > 1$, and hence the denominator of (2) becomes very large, leading to a population crash. Thus, for boom-bust dynamics two things are needed: λ should not be much larger than 1, so that exponential growth from low densities is long and slow, and β should be large, so that population crashes

are severe. Note that the fact that λ appears in the denominator is simply because we want to have the carrying capacity as an explicit parameter in the model (i.e., in the term $N(t)/K$). The appearance of λ in the denominator of (2) has no bearing on the qualitative requirements for boom-bust dynamics, which are simply that λ is not much larger than 1 and β is large.

We note that, as pointed out in the manuscript, large β corresponds to scramble competition. To see this, one may plot the per capita number of offspring, which is the right hand side of (2) divided by $N(t)$, as a function of the population size $N(t)$. For large β , such a plot qualitatively always looks the same, regardless of the value of λ : it is essentially flat for $N(t) < K$, and then falls abruptly to very small values as N_t crosses the carrying capacity K . Such a shape of the per capita number of offspring as a function of population density is the hallmark of scramble competition. All these properties of discrete-time models, including the fact that large β correspond to scramble competition, are very well known from the large literature on discrete-time models.

That the parameter β rather than λ better characterizes the strength of competition can also be seen from the following argument: For any β in the range that we consider, the boom phase consists of almost exponential growth for all populations, illustrated e.g. in Fig. 4. The timing of the crash does not depend on β or on λ , but rather on the phenotypic composition of the evolving community. However, the magnitude of the crash increases exponentially with β , but not with λ . We also note that the intrinsic growth rate cannot be rescaled into the definition of time in discrete-time models, because there is no explicit time in discrete maps. Instead, such maps usually link population size at the next time unit (often corresponding to a generation) to the current population size.

Finally, we note that there are many other models that can produce such boom-bust patterns but that are, in our opinion, mathematically less elegant. In the revised manuscript we briefly discuss such a model with pure exponential growth and busts implemented “by hand” when the effective population density experienced by a given phenotypic species exceeds the carrying capacity. Such a model yields the same increase in diversity as (4). A discussion of intricacies of parametrization of model (4) and its complete phase space would in our opinion only distract a reader from our main message, rather than adding anything to it (see also our response to reviewer 1). We changed the wording in the text where we introduce the model (1-2) and (4), stating that our prime interest in this model is its ability to produce boom-bust dynamics.

Changes made:

- We significantly changed the first paragraph of the Model and simulation methods section (see our reply to point 4. below).
- We added the following sentence to the second paragraph of the Model section:

This is reflected by the shape of the per capita number of offspring as a function of

population size, given by the right hand side of (1) when divided by $N(t)$, and viewed as a function of $N(t)$: for any λ , and for high β , the per capita number of offspring is approximately constant until the population size reaches the vicinity of K , but as N increases above K the of offspring falls rapidly to very low values, essentially generating a population crash as soon as the population size is above K .

3. Another issue I found confusing is that there is not enough literature as it should be. Biodiversity and paradox of plankton have been a big topic in ecology and there has been a huge amount of proposed models and mechanisms, including the effect of demographic fluctuations, evolutionary traits, spatial inhomogeneity etc. For example, a paper by Gavina et al, Sci Rep. 8(1), 1198 (2018) considering the higher order of interspecies competition shows that the coexistence can be sustained by weak interspecies competition from the crowding effect, which seems to opposite to the authors' claim. Also the phenomenon similar to the asynchronous "boom-bust" dynamics due to evolution and stochasticity is studied in the paper by Xue and Goldenfeld, Phys. Rev. Lett. 119, 268101 (2017). I suggest the authors to include more literature in the introduction and indicate how the authors model is different from them or justify why the authors choose the feature in their model.

Reply: We included a paragraph in the Discussion referencing these papers. However, there is not much overlap between the models included in these references and our model. The first papers considers a modification of continuous-time competition model due to crowding effects, which can generate slightly increased levels of diversity, but apparently by far not the levels of diversity reported in our paper. The second reference considers a modification of a predator-prey system, which is an entirely different group of models with its own mechanisms for the maintenance of diversity in both predators and prey. In particular, in contrast to our consumer-intrinsic mechanism of competition, the kill-the-winner mechanism envisaged in the second reference can be regarded as extrinsic to the consumer (prey) population.

Changes made:

- We added the following paragraph towards the end of the Discussion section:

We speculate that the mechanisms and results reported here are not limited to competition models, but could also be manifest in communities with other ecological interactions, e.g. in communities with crowding effects (Gavina et al 2018), or in communities containing both predators and prey (Xue and Goldenfeld 2018). Whenever the population dynamics exhibit patterns of rapid growth interspersed by crashes (as may e.g. be expected in many predator-prey systems), temporal desynchronization can occur spontaneously and thus lead to increased diversity. Such effects were shown in (Xue and Goldenfeld 2018), who reported that "kill-the-winner" mechanism, in which

predation generates crashed in the most abundant consumer species, can generate increased levels of diversity. These mechanism differ from the ones reported here in that they are extrinsic to the crashing consumer species (and it is difficult to compare those system to baseline systems with stable ecological dynamics).

4. Below Eq (2), it is not clear to me why λ has to be larger than 1 to avoid extinction. I suggest that the authors explain more how this is obtained.

Reply: We thank the Reviewer for asking to clarify this point. In fact, the formulation we gave as eq. (1) in the original submission (and is now eq. (2) i the revised version) is only valid for $\lambda > 1$, and so our statement was confusing. We now explain that the general form of the model we use is $N(t + 1) = N(t)\lambda/(1 + aN(t)^\beta)$, where $\lambda > 0$, $a > 0$ and $\beta > 0$ are parameters. In this formulation, it is clear that $\lambda < 1$ leads to extinction (and $N = 0$ is the only steady state). On the other hand, if $\lambda > 1$, there is an additional steady state at $K = ((\lambda - 1)/a)^{1/\beta}$, which is the carrying capacity of the population. In this case (i.e., if $\lambda > 1$ so that $K > 0$ exists), we can reformulate the model as eq. (1) in the original text, which became eq. (2) in the revised version. So eq. (2) in the revised version should be viewed as the version of the model that we use throughout, assuming $\lambda > 1$.

Changes made:

- We changed the first paragraph of the Model and simulation methods section to the following:

To accommodate various types of ecological dynamics, we consider ecological models given by difference equations, and hence set in discrete time. The basic ecological model we use is a difference equation that links population densities N of two consecutive generations t and $t + 1$,

$$N(t + 1) = F(N(t)) = N(t) \frac{\lambda}{1 + aN(t)^\beta}. \quad (1)$$

where $\lambda > 0$ is the per capita number of offspring, and $a > 0$ and $\beta > 0$ are parameters describing the effect of competition. For $\lambda < 1$, $N(t)$ converges to 0 for any initial condition $N(0) > 0$, and hence extinction is the only possible outcome. We therefore assume $\lambda > 1$ in what follows. In that case, model (1) has a non-zero equilibrium at $K = ((\lambda - 1)/a)^{1/\beta}$, which is the carrying capacity of the population. It is then convenient to write (1) as:

$$N(t + 1) = N(t) \frac{\lambda}{1 + (\lambda - 1)(N(t)/K)^\beta}, \quad (2)$$

as this makes it easy to formulate the model in terms of continuous phenotypes (see below). Model (2) was shown to fit well a wide range of data, and for $\beta = 1$ can be derived from the logistic differential equation by integration over a finite time interval.

5. *Following Comment 4, it is not clear to me why the population dynamics converges to a steady state if and only if $|1 - ((\lambda - 1)/\lambda)\beta| < 1$. Do you mean $|1 - ((\lambda - 1)/\lambda)\beta| < 0$ instead? I suggest that the authors briefly explain this or list the reference for it.*

Reply: The condition we give is the well-known, universally valid criterium for linear stability of an equilibrium point of a discrete-time map: the absolute value of the derivative of the right hand side of (2), evaluated at the equilibrium point, has to be < 1 . The corresponding condition in continuous time is that the corresponding derivative, evaluated at the corresponding equilibrium point, has to be < 0 . Technically, these criteria follow from a Taylor expansion of the relevant maps and assess whether a small perturbation away from the equilibrium point increases or decreases over time. We are guessing that this question arose because the Reviewer assumed that we consider continuous-time dynamical systems (i.e., with the derivative $dN/dt = F(N)$ instead of the next generation population density $N(t + 1) = F(N(t))$ on the left-hand side of the dynamic equation.) This is, however, not the case.

6. *Higher diversity in higher dimension seems to be straightforward to me, with or without competition, because the phenotype space is larger. I suggest the authors to illustrate more about the purpose of studying higher dimension.*

Reply: This may be true for the “baseline diversity” that evolves in communities with stable equilibrium ecological dynamics. We systematically studied this effect in Doebeli, M. and Ispolatov, I. (*American Naturalist*, 2017). However, the effect of dimension is far from obvious in the present model: in higher phenotypic dimensions the number of phenotypically similar species (or nearest neighbours in phenotype space) increases linearly with dimension, and it may in fact be expected that this makes it harder to desynchronize the boom-bust population dynamics between different species, as is required for enhanced diversity. Thus, one could expect the effect to weaken with higher dimensions of phenotype space, but our results actually show that this is not the case.

Changes made:

- We added the following sentence at the end of the 3rd paragraph of the Discussion section:

We note that this latter result is not obvious, as with higher phenotypic dimensions the number of phenotypically similar species (nearest neighbours in phenotype space) increases linearly with the dimension, which may be expected to make de-synchronization of neighbouring boom-bust cycles more difficult due to denser phenotype packing.

7. In the authors' model the growth rate is independent of phenotype and there is no trade-off. It seems unnatural and I suggest the authors to justify this assumption.

Reply: None of the phenotypic dimensions that we consider in our models is related to the growth rate, which is in line with many other classic competition models. Studying such tradeoffs was not on our agenda, and there was no need to introduce any tradeoffs to address the effects of boom-bust dynamics on diversity. Of course, one can always complicate any model by introducing additional assumptions, such as tradeoffs, which would lead to new insights. However, our model is purely about competitive effects, and tradeoffs between growth rate and other model parameters did not seem relevant in this context. Perhaps future work will reveal interesting insights with regard to such tradeoffs in these models.

Reviewers' comments:

Reviewer #2 (Remarks to the Author):

Major comment: The authors have quite responded to most of my comments and modified the manuscript accordingly. However, it looks like that the authors misunderstood my question about the derivation of their model and therefore the authors seemed to excessively celebrate discrete time models. My question is never whether discrete time models are more basic or fundamental than continuous time models for densities. Actually I am never a fan of continuous time models or continuous density equations. My question is simply how the model the authors started from, i.e. Eq (1) for general β can be derived. And I do not agree with the authors in their response that "any continuous-time model is in the end also derived in an ad hoc manner. For example, the logistic continuous-time model is obtained by simply assuming that the growth rate declines linearly with density", because one can analytically derive the logistic growth by taking the mean-field average of the stochastic dynamics of the chemical reactions, and we know under which conditions and limits those approximations could work or fail. This is by no means ad hoc. The references I mentioned in my previous report are just some examples that I found, and I can understand how those equations could be analytically derived and therefore I know what their results were based on, so that is why I was asking if the authors can also explain how they derive the equation which they are simulating (by the way I want to emphasize that I am none of the authors of those references I mentioned, and I expected that the authors should do a more comprehensive literature search and compare different mechanisms about the boom-bust phenomenon, since there have been many studies on exclusive competition versus diversity as I mentioned previously). On the contrary, I cannot understand the authors' model with the similar manner. Since the model that the authors are using is an empirical one and was adopted from the other studies, I do not think the authors should assume that the readers are supposed to take it for granted. Especially the phase diagram and parameter range given by the model could be non-realistic as the authors admitted, which could restrict the validity of their model and results, and the readers should be aware of that. So I request that the authors clearly state at the beginning of Eq (1) that the basic model with general β is an empirical model which is adopted from the literature and is not derivable analytically.

Reply to reviewer comments on "Boom-bust population dynamics can increase diversity in evolving competitive communities", submitted to *Communications Biology*

February, 22 2021

Reviewer 2:

We thank the reviewer for acknowledging that we have constructively responded to almost all the points that the reviewers have raised. It seems that there is only one issue remaining, which concerns the nature of our basic model given by equation (1). Specifically, after giving further arguments to make their point, the reviewer requests that: *"So I request that the authors clearly state at the beginning of Eq (1) that the basic model with general β is an empirical model which is adopted from the literature and is not derivable analytically."*

We agree that model (1) cannot be derived from first principles, as has also been argued in ref 23. In that sense, the model cannot be derived analytically. We note, however, that it is straightforward to come up with an individual-based, stochastic model whose mean-field limit is model (1). This can e.g. be done by viewing the denominator of (1) as the inverse of a survival probability. One then starts with a finite number of individuals, each of which survives probabilistically according to this survival probability (which decreases with increasing numbers of individuals due to competition), and then each of the surviving individuals has e.g. a Poisson distributed number of offspring, with λ being the mean of that Poisson distribution. Model (1) is then the mean-field average of this stochastic process, just like the logistic equation is the mean field average of an underlying stochastic, individual-based process.

Nevertheless, we followed the request of reviewer 2 by inserting the following sentences in the paragraph following eq. (2). We hope that this settles the outstanding issue identified by reviewer 2.

Added text: However, it is important to note that the model given by (1) and (2) is phenomenological in nature. While other simple discrete-time models can be derived from first principles, this does not appear to be the case for model (1) (see ref (23)). Rather, this model should be viewed as an empirical model that can exhibit a wide array of dynamic regimes, including the boom-bust regime that will be of paramount importance in this paper (see below).